# Spiculogenesis and biomineralization in early sponge animals

Qing Tang [1], Bin Wan[2,3], Xunlai Yuan[2,3], A.D. Muscente[4] & Shuhai Xiao [1]

Most sponges have biomineralized spicules. Molecular clocks indicate sponge classes diverged in the Cryogenian, but the oldest spicules are Cambrian in age. Therefore, sponges either evolved spiculogenesis long after their divergences or Precambrian spicules were not amenable to fossilization. The former hypothesis predicts independent origins of spicules among sponge classes and presence of transitional forms with weakly biomineralized spicules, but this prediction has not been tested using paleontological data. Here, we report an early Cambrian sponge that, like several other early Paleozoic sponges, had weakly biomineralized and hexactine-based siliceous spicules with large axial filaments and high organic proportions. This material, along with Ediacaran microfossils containing putative non-biomineralized axial filaments, suggests that Precambrian sponges may have had weakly biomineralized spicules or lacked them altogether, hence their poor record. This work provides a new search image for Precambrian sponge fossils, which are critical to resolving the origin of sponge spiculogenesis and biomineralization.

---

[1] Department of Geosciences and Global Change Center, Virginia Tech, Blacksburg, VA 24061, USA. [2] State Key Laboratory of Palaeobiology and Stratigraphy, Nanjing Institute of Geology and Palaeontology and Center for Excellence in Life and Palaeoenvironment, Chinese Academy of Sciences, 210008 Nanjing, China. [3] University of Chinese Academy of Sciences, 100049 Beijing, China. [4] Department of Geological Sciences, University of Texas, Austin, TX 78712, USA. Correspondence and requests for materials should be addressed to S.X. (email: xiao@vt.edu)

Sponge animals, either paraphyletic at the base of the animal tree[1] or forming a monophyletic clade that is the sister group of all other animals[2–4], likely diverged in the Precambrian. Biomarker fossils suggest that sponge classes diverged no later than the Cryogenian Period[5,6], although their interpretations remain a matter of debate[7–10]. Molecular clock studies[11], including recent ones with improved taxonomic sampling and independent of the aforementioned biomarkers as a calibration[12,13], point to a similar antiquity of sponge classes. More importantly, the presence of bilaterian animals in the Ediacaran Period[14–16] strongly indicates the divergence of sponges and even sponge classes in the Precambrian, particularly if sponges are paraphyletic[1]. It has been postulated that the last common ancestor of sponges (or that of silicean sponges) may have had biomineralized spicules[1,8], yet the Precambrian fossil record of biomineralized sponge spicules is ambiguous at best[17,18]. Previous attempts to resolve this 'missing glass problem' were focused on taphonomic factors that might have limited the preservation of sponge spicules in the Precambrian[1]. However, emerging phylogenetic data do not require the presence of biomineralized spicules in the last common ancestor of demosponges (and that of siliceans)[12,19,20], prompting an alternative hypothesis that spicules may have evolved independently among sponge classes and perhaps long after the divergence of sponges[21]. This hypothesis predicts that the last common ancestor of sponges was aspiculate and that transitional forms with weakly biomineralized spicules characterize early sponges. Thus far, to our knowledge, there have been no paleontological data to test this predication.

Here we report a new sponge fossil, *Vasispongia delicata* Tang and Xiao, n. gen. & sp., from the early Cambrian (Age 2) Hetang Formation in South China (Supplementary Fig. 1). *V. delicata* has hexactine-based siliceous spicules with a large axial filament and a high organic content. We hypothesize that early sponges may have had weakly biomineralized spicules with low fossilization potential. Thus, although sponges or sponge classes may have diverged in the Precambrian, biomineralized spicules may have evolved later and independently among sponge clades.

## Results

**Phylum Porifera Grant, 1836 (ref. [22].)**

**Class, Order, Family uncertain**

**Genus *Vasispongia* Tang and Xiao gen. nov.**

**Type species**. *Vasispongia delicata* Tang and Xiao sp. nov.

**Diagnosis**. As for type species.

**Occurrence**. Specimens were recovered from the stone coal unit of the Hetang Formation (Stage 2, lower Cambrian) in the Lantian area of Anhui Province, South China.

**Etymology**. Genus name derived from Latin *vas*, referring to the vase-shaped morphology of the sponge body.

**Remarks**. *Vasispongia* sp. nov. can be distinguishable from other Paleozoic sponges by its hexactine-based spicules characterized by a cylindrical axial filament and an outer organic layer which together account for large proportion (ca. 10–100%) of spicule diameter.

### *Vasispongia delicata* Tang and Xiao sp. nov.

(Figures 1–4; Supplementary Figs. 2 and 4)

**Holotype**. VPIGM-4699 in Fig. 1a, reposited at Virginia Polytechnic Institute Geosciences Museum.

**Paratype**. VPIGM-4726 in Supplementary Fig. 2b, reposited at Virginia Polytechnic Institute Geosciences Museum.

**Diagnosis**. Small vase-shaped sponge with weakly biomineralized skeleton composed of densely arranged hexactine-based spicules that consists of a large proportion (ca. 10–100% in diameter) of organic material, including a cylindrical axial filament and an outer organic layer. A thin silica layer may be present between the axial filament and the outer organic layer.

**Occurrence**. The stone coal unit of the lower Hetang Formation (Stage 2, Lower Cambrian), Lantian area, South China.

**Etymology**. Species epithet derived from Latin, *delicatus*, with reference to the delicately preserved and weakly mineralized spicules.

**Materials**. Nineteen specimens were recovered from the Hetang Formation.

**Remarks**. Spicule microstructures of *Vasispongia delicata* n. gen & sp. are broadly similar to the siliceous sponge *Lenica* sp. of the early Cambrian Hetang Formation[23] and *Cyathophycus loydelli* of the Ordovician Llanfawr Mudstones Formation[24], which also develop clear organic components in their spicules. However, *Vasispongia delicata* is distinct from those Paleozoic siliceous sponges by a combined feature of a cylindrical axial filament and an organic outer layer that account for a large proportion (up to 100% in diameter) of the spicules.

## Description

In total, 19 well-preserved specimens of *V. delicata* were collected from organic-rich mudstones of the lower Hetang Formation. These fossils are millimeters in size (Supplementary Data 1) and are preserved as discoidal or elliptical carbonaceous compressions. In some specimens, the disc or ellipsoid has a neck-like extension and a central cavity (Fig. 1a–c and Supplementary Fig. 2), which are interpreted as possible osculum and spongocoel, respectively. They each contain abundant monaxons, diaxons, and triaxons that are randomly distributed in the organic remains. Most spicules are non-mineralized or partially demineralized through taphonomic processes[25], resulting in cylindrical external molds (Fig. 1d, e). These molds are micrometers in diameter (Supplementary Data 1) and typically contain an organic cylindrical structure located centrally (Fig. 1d) or eccentrically due to secondary dislocation (Fig. 1e). The cylindrical structures are straight or slightly sinuous (Fig. 1h, i), and some are incompletely preserved or fragmented due to degradation (Fig. 1e–g). Although some cylindrical structures may be monaxons, others are clearly stauractines (Fig. 1f), pentactines (Fig. 1h), and hexactines with orthogonal rays (Fig. 1i, j). The overall shape of the cylindrical structures is consistent with the hosting spicules.

The organic cylindrical structures are variable in diameter, accounting for ~10% to nearly 100% of the spicule diameter (Supplementary Fig. 3a and Supplementary Data 1). Some of them consist of two parts, a solid inner core and a concentric outer lamella (Figs. 1g and 2a–c). The inner core also has hexactine-based rays, which may taper distally (Fig. 1h, i) or be aborted to form a short protuberance (arrows in Fig. 1h, j and Supplementary Fig. 4a, b). The core is circular in cross section (Figs. 2a–c, 3a, b, and 4a), accounting for much of the cylindrical

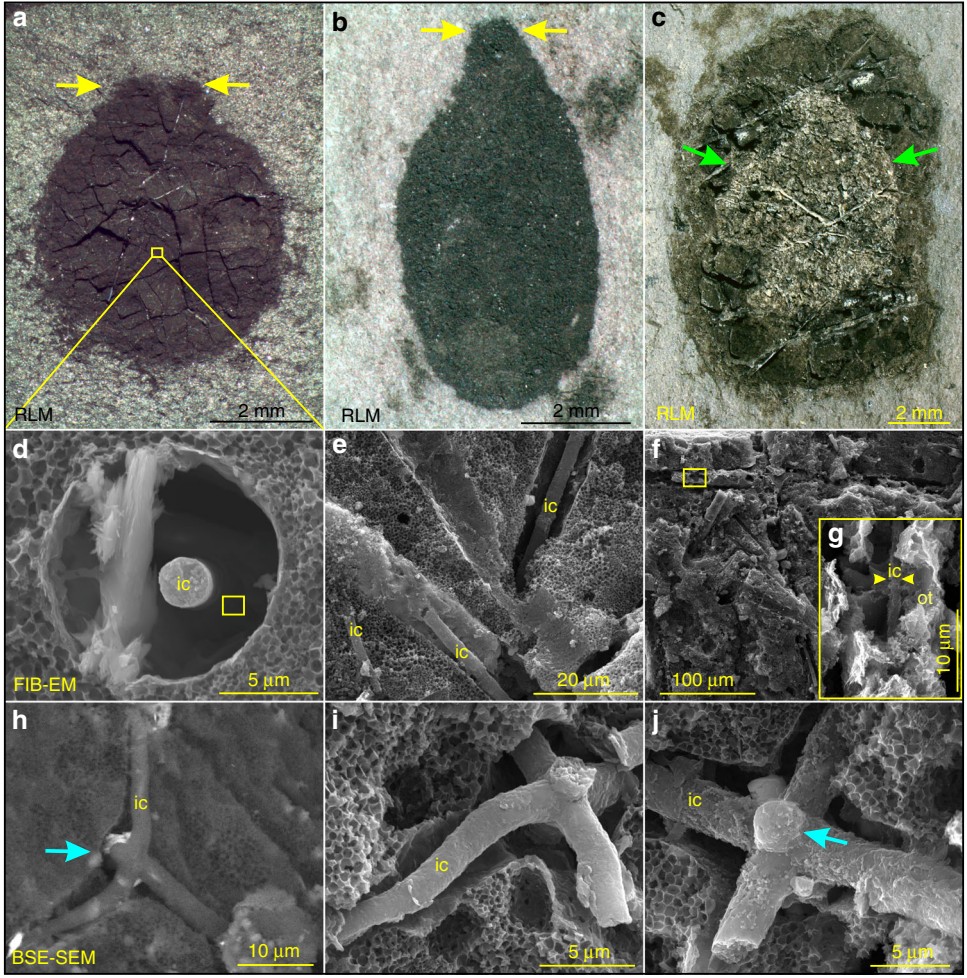

**Fig. 1** *Vasispongia delicata* Tang and Xiao, n. gen. & sp. from the Hetang Formation. **a–c** Carbonaceous compressions of sponge body fossils. Yellow and green arrows bracket putative osculum and spongocoel, respectively. Polygonal cracks in **a** are manifested as black in color. VPIGM-4699 (16-HT-T6-1-1), VPIGM-4700 (16-HT-T5-6-1), and VPIGM-4701 (16-HT-T3-77-1), respectively. **d–j** Demineralized spicules. **d** and **g** are magnifications of rectangles in **a** and **f**, respectively. Blue arrows in **h** and **j** point to protuberances or aborted rays. **e**, **f** VPIGM-4702 and VPIGM-4703, respectively; **h–j** VPIGM-4704, VPIGM-4705, and VPIGM-4706, respectively. Honeycomb-like structures (**d**, **e**, **i**, **j**) in the carbonaceous matrix are molds of framboidal pyrite. **a–c** are reflected light micrographs (RLM) and **h** is backscattered electron scanning electron microscopy (BSE-SEM) micrograph. All other images in this and other figures are secondary electron scanning electron microscopy (SE-SEM) micrographs unless otherwise noted. ic inner core, ot outer lamella

structure and sometimes being the only component of the cylindrical structure when the outer lamella is not developed or preserved (Figs. 1d, e, h–j and 2e, f). The core consists of elongate nanoparticles loosely compacted with nanoporous structures, which can be observed on both transverse sections (Fig. 2c, d) and on the surface (Fig. 2e). The surface of the core is sometimes ornamented with ridges (Fig. 2e–g and Supplementary Fig. 4c–e) and tubercles (Fig. 2h and Supplementary Fig. 4e, f). The ridges are longitudinally or obliquely oriented, with an angle of 0–57° relative to the longitudinal axis of the core. They are broadly similar to the ridges on the core or middle layer of some spicules of the Cambrian sponge *Lenica* sp. from the Hetang Formation (see Figs. 1b and 2a in ref. [23]).

The outer lamella of the cylindrical structure is a concentric layer enclosing the inner core (Figs. 2a–c, 3, and 4a). It is typically coarse and uneven on its outer surface (Figs. 2a and 3e), and consists of porous amorphous nanoparticles (Fig. 2d). The gaps between the inner core and outer lamella and between the outer lamella and the matrix are variable (Fig. 2a–c, 3e, i and 4a). In some specimens, the inner core is tightly enveloped by the outer lamella, and the entire spicule is fully occupied by the cylindrical structure without any appreciable gaps (Fig. 3a, b). A few

longitudinally exposed specimens show variable preservation: segments of the cylindrical structure are well preserved with a very narrow gap (Fig. 3c, d, f–h) whereas the rest of the cylindrical structure has wider gaps because the outer lamella is lost (Fig. 3e, f, i).

Energy dispersive X-ray spectroscopy (EDS) analyses indicate the cylindrical structures, including the inner core and outer lamella, are mainly composed of organic carbon (Fig. 4). EDS point analyses show that the inner core and outer lamella have higher C peaks but lower Al and Si peaks than the surrounding matrix (Fig. 4a, b). EDS elemental maps (Fig. 4b) confirm the organic nature of the cylindrical structures. In addition, a few cylindrical structures consist of an inner core rich in organic carbon and an outer siliceous layer rich in Si and O (Fig. 4c, d). This outer siliceous layer may represent the remnant of siliceous spicule that survived diagenetic demineralization[25].

### Preservation

Fossils of *V. delicata* are preserved in organic-rich mudstones of the lower Hetang Formation at Lantian in southern Anhui Province, South China. This unit of organic-rich mudstone is sometimes called the stone coal unit[26] or the anthracite layer[27]

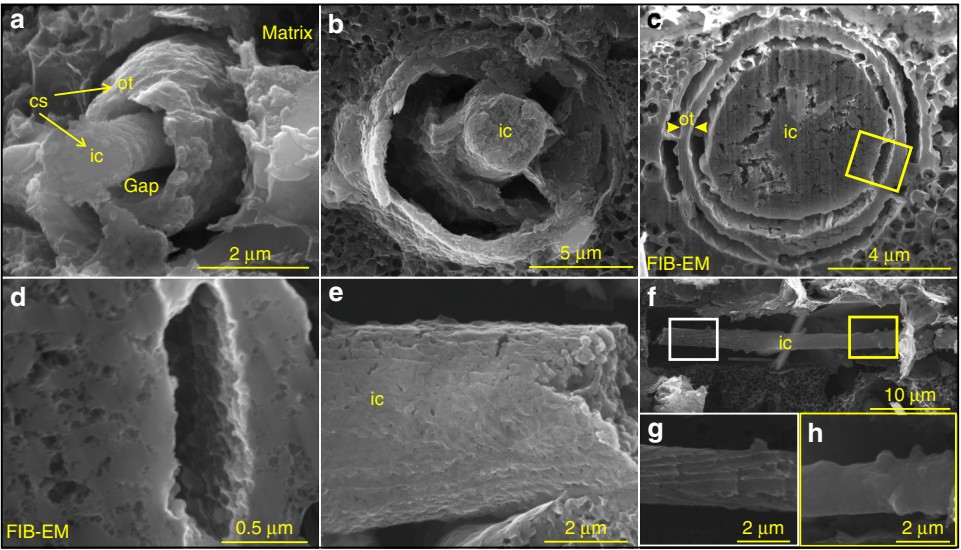

**Fig. 2** Demineralized spicules of *Vasispongia delicata*. **a–d** Cross-sectional views of spicules with inner core and outer lamella. VPIGM-4707, VPIGM-4708, and VPIGM-4709, respectively. **d** is a magnification of the rectangle in **c**. **e**, **f** Lateral views of axial filaments. VPIGM-4710 and VPIGM-4711, respectively. **g**, **h** Magnifications of white and yellow rectangles in **f**, respectively. cs cylindrical structure, ot outer lamella, ic inner core

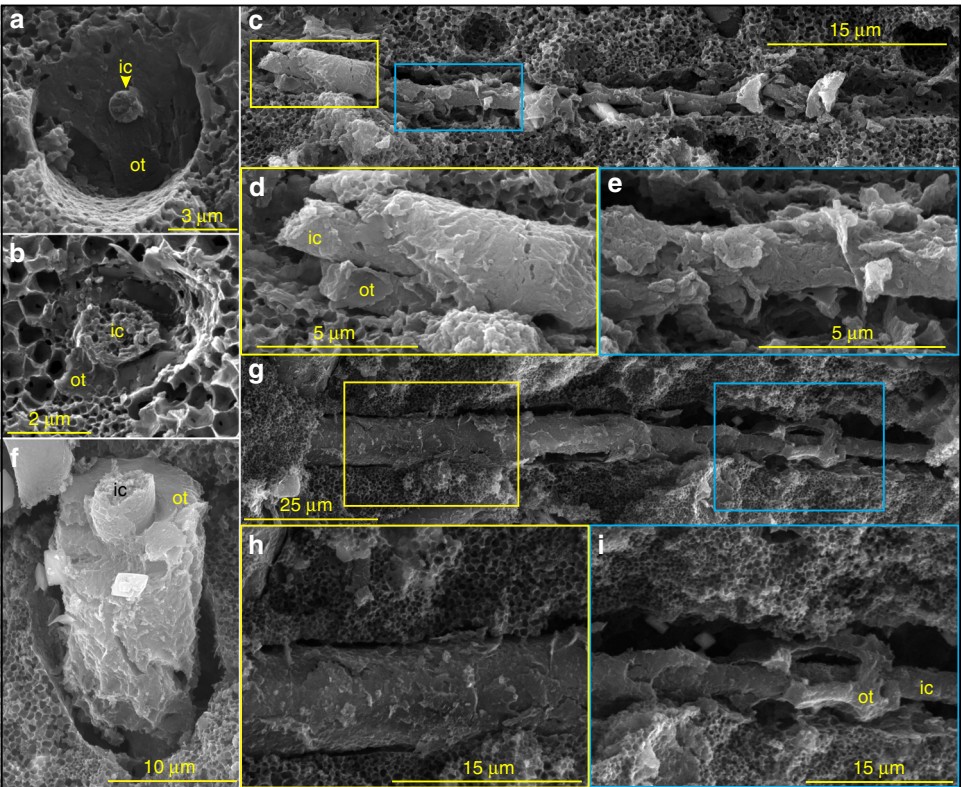

**Fig. 3** Cylindrical structures of *Vasispongia delicata*, showing concentrically arranged inner core and outer lamella. **a**, **b** Cross-sectional views. VPIGM-4712 and VPIGM-4713, respectively. **c**, **f**, **g** Lateral views. VPIGM-4714, VPIGM-4715, and VPIGM-4716, respectively. **d**, **h** Magnifications of the yellow frames in **c** and **g**, respectively, showing well-preserved outer lamellae. **e**, **i** Magnifications of the blue frames in **c** and **g**, respectively, showing partially degraded outer lamellae. ic inner core, ot outer lamella. Honeycomb-like structures (**a–d**, **f–i**) in the carbonaceous matrix are molds of framboidal pyrite

(note that the basal boundary of the Hetang Formation was misplaced in ref. [27]. and should be placed beneath the chert unit just below the anthracite layer, ~450 m). The stone coal unit contains up to ~15 wt% total organic carbon content[27] and locally abundant framboidal pyrite[28] (e.g., honeycomb-like structures in Fig. 1d–e are molds of framboidal pyrite). Iron speciation data, redox-sensitive metal concentrations, and framboidal pyrite size distributions indicate that this unit was deposited in quiet and anoxic (ferruginous or sulfidic) environments[27–29], although brief episodes of dysoxic to oxic condition may have occurred[28]. Such quiet and anoxic conditions, as well the fine-grained clay minerals in the stone coal unit, may have contributed to the preservation of soft-bodied animals and articulated sponge body fossils in the stone coal unit[26,30].

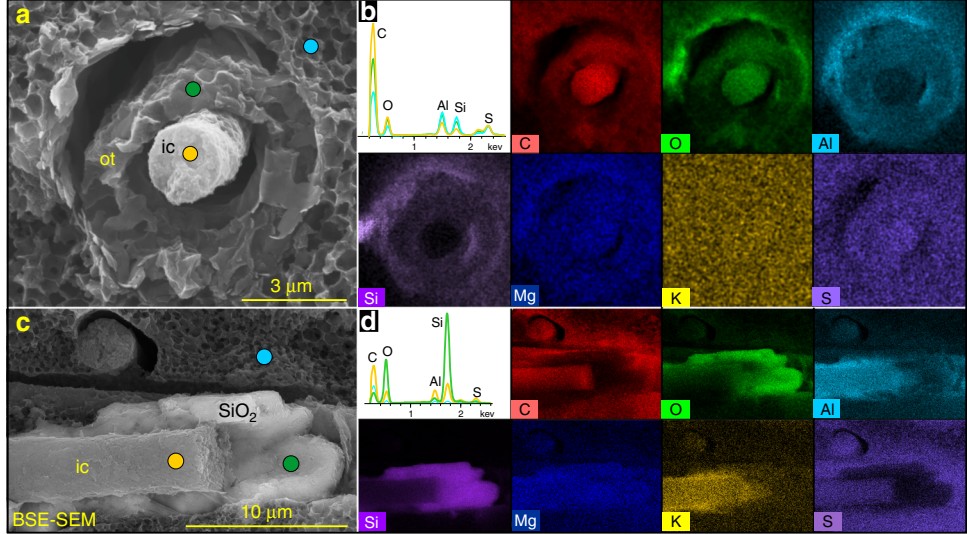

**Fig. 4** Preservation of organic and biosilica structures in *Vasispongia delicata*. **a** Demineralized spicule. VPIGM-4717. **b** EDS point analysis and element maps of **a**. **c** BSE-SEM micrograph of partially demineralized spicule. VPIGM-4718. **d** EDS point analysis and element maps of **c**, showing organic axial filament enveloped by a silica lamella. Colored dots in **a** and **c** denote the location of EDS point analyses shown in **b** and **d**, respectively. ic inner core, ot outer lamella, SiO₂ siliceous layer

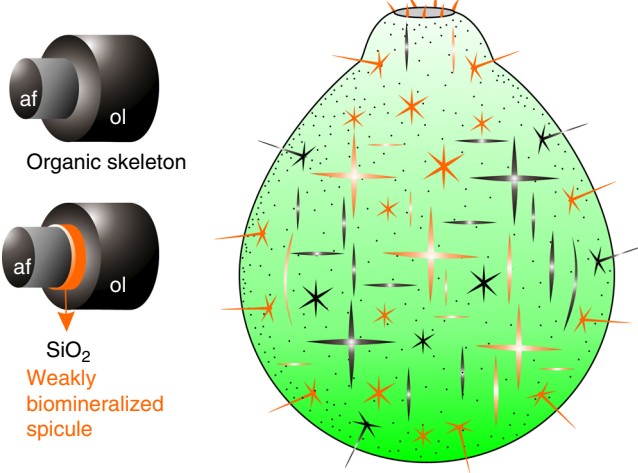

**Fig. 5** Spiculogenesis and morphological reconstruction of early sponges. Schematic reconstructions of early sponges with weakly biomineralized spicules and entirely organic skeletons as inferred from *Vasispongia delicata*. Pattern of spicule/skeleton distribution and orientation is conjectural but based on Cambrian reticulosan sponges[8]. Organic skeletons and weakly biomineralized spicules in the sponge body reconstruction are colored in black and orange, respectively. af axial filament, ol organic layer, SiO₂ siliceous layer

Several features of *V. delicata* are likely of taphonomic origin. As pointed out above, the moldic preservation of spicules (Fig. 1d–j) is related to taphonomic demineralization or the dissolution of biomineralized structures, a phenomenon that particularly common in organic-rich sediments[25]. By the same token, the gaps between the inner core and outer lamella (Figs. 2a–c and 4a), as well as those between the outer lamella and the surrounding carbonaceous matrix (Figs. 2a, c and 4a), are likely results of taphonomic demineralization. This interpretation is supported by the presence of residual silica in incompletely demineralized spicules (Fig. 4c). Finally, the eccentric dislocation (Fig. 1e) and fragmentation of some cylindrical structure (Fig. 1d, i, j), as well as the ripped outer lamellae (Fig. 3e, i), are also taphonomic in origin.

Despite these taphonomic features, *V. delicata* preserves genuine biological structures. The consistent presence of an osculum and a spongocoel in multiple specimens (Fig. 1a–c; Supplementary Fig. 2) indicates that these structures are biological structures rather than preservational artifacts. Of course, oscula and spongocoels are not preserved in all specimens, because their preservation is dependent on lateral compression and exfoliation of the sponge body wall, respectively. More importantly, the organic cylindrical structure in the center of the spicules is of biological origin. This interpretation is supported by the observation that the cylindrical structure is consistently composed of an inner core surrounded by an outer lamella (Figs. 2a–c, 3a–c, f, and 4a) or outer siliceous layer (Fig. 4c), and that the inner core is ornamented with exquisitely preserved ridges and tubercles (Fig. 1e–g and Supplementary Fig. 4). Given that the Cambrian sponge fossil record is dominated by disarticulated spicules with few microstructures, *V. delicata* is considered an exceptionally preserved body fossil with exquisitely preserved organic component of the spicules.

## Interpretation

The presence of an osculum, a spongocoel, and hexactine-based spicules unambiguously indicates that *V. delicata* is a crown-group sponge animal (if sponge is a monophyletic clade), as the combination of these features is present only in sponges and not in other animals. The inner core and outer lamella are interpreted as the axial filament and concentric organic layer, respectively, of *V. delicata* spicules. Their concentric arrangement is consistent with the appositional growth of extant sponge spicules[31]. The carbonaceous composition, size, and central location of the inner core are also consistent with an axial filament interpretation. Finally, the nanoporous and nanoparticle structures in the inner core are similar to those in the axial filaments of extant silicean sponges[32], for example, *Suberites domuncula* (see Fig. 2l in ref.[31]).

The interpretation of the outer lamella in *V. delicata* spicules as a concentric organic layer surrounding the axial filament is supported by similarities with modern and fossil sponges. An interpretive analog is found among extant silicean spicules that develop organic layers intercalated with silica lamellae[31]. Similar

to the axial filament, these organic layers are predominately composed of silicatein-collagen complex and play important roles in mediating the formation of biosilica lamellae[31]. Concentric organic layers have also been reported from the Ordovician silicean *Cyathophycus loydelli*, whose spicules typically consist of intercalated silica lamellae and organic layers[24].

We interpret that spicules of *V. delicata* may have originally had a biomineral layer between the axial filament and outer organic layer (Fig. 5), and the biomineralic layer was subsequently demineralized, resulting in an empty space or a gap. This biomineralic layer likely consisted of biosilica, given the preservation of a siliceous layer surrounding the axial filament of partially demineralized spicules (Fig. 4c) and the fact that extant hexactine-based spicules are always siliceous. The gap between the outer lamella and surrounding carbonaceous matrix may represent another demineralized layer of biosilica.

## Discussion

The axial filaments of *V. delicata* stand out in their cylindrical shape. The cross-sectional symmetry of axial filaments is an important character for class-level identification of extant sponges, with hexactinellids characterized by quadrangular axial filaments and demosponges by triangular or hexagonal ones[33]. However, axial filaments are extremely rare in the fossil record. Some spicule fossils preserve a cylindrical axial canal[24], but this is an unreliable morphological proxy for the axial filament[23,24], because it can be reshaped and enlarged by mineral dissolution[34]. Organically preserved axial filaments, on the other hand, provide more reliable information about their biological precursors. Cylindrical axial filaments in *V. delicata* and other Cambrian-Ordovician siliceans, such as *Cyathophycus loydelli* (Fig. 3i of ref. [24].) and *Lenica* (Fig. 5 in ref. [23].), indicate that cylindrical axial filaments existed in multiple early Paleozoic sponges.

The axial filaments *V. delicata* are also distinguished from their modern counterparts by their relatively larger but variable diameters as well as by the relatively thick organic layers that surround them. Overall, the axial filaments are 0.7–9.4 μm in diameter (representing 6–94% of spicule diameter) and the surrounding organic layers are 0.2–3.8 μm in thickness (Supplementary Data 1). The variation in axial filament diameter and organic layer thickness may be partly due to taphonomic degradation; given that thermal maturation, devolatilization, and chemical oxidation of organic structures tend to result in volume reduction, the upper end of the range of measurements is more likely to approach the original sizes. For comparison, axial filament diameter and organic layer thickness of modern silicean spicules are 0.1–2 μm (or 3–6% of spicule diameter) and 2−10 nm (e.g., in the hexactinellid *Euplectella* of ref. [35].), respectively. There is also a positive relationship between spicule diameter and axial filament diameter of *V. delicata* (Supplementary Fig. 3b), suggesting that the axial filaments of *V. delicata* may have grown in size as spicules matured. Such growth may have been accommodated by skeletal remodeling and indicates that the axial filaments may have functioned as an important structural component of the spicules.

A high organic content has also been observed in some other Paleozoic sponge spicules. A compilation of fossil and extant sponge spicules shows that the organic proportion of *V. delicata* spicules and several other early Paleozoic sponge spicules is much higher than those of younger ones, despite that the latter values are generally maximum estimates based on axial canal measurements (Fig. 6 and Supplementary Data 2). Indeed, some Cambrian sponges, such as *Vauxia*[36], have organic fibrous skeletons, whereas post-Ordovician and extant sponge spicules show

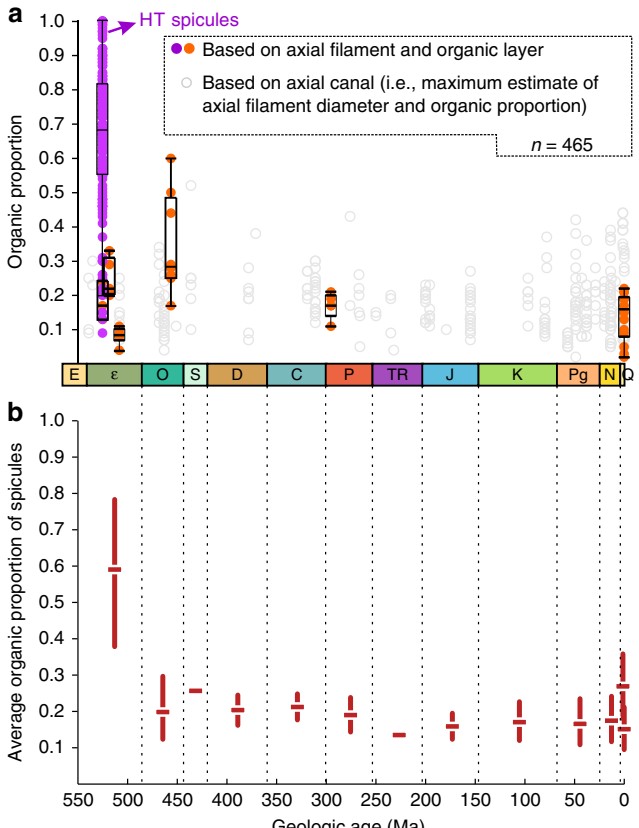

**Fig. 6** Relative organic proportion in fossil and extant sponge spicules. **a** Plot of organic proportion measurements of fossil and extant sponge spicules. Organic proportion of sponge spicules was estimated as the relative thickness of combined organic structures (including axial filament, organic layer, and outer sheath) in spicules. When axial filaments are not preserved, axial canals were measured as a maximum estimates of axial filament size. Data based on axial filament measurements are plotted with box and whiskers, where box represents 25–75th percentiles, bar within box represents 50th precentile, and whiskers represent 1.5 × IQR (interquartile range) extensions from the box up to minimum and maximum measurements, with outliers plotted beyond whiskers. See Supplementary Data 1 and 2 for data. **b** Mean and 95% confidence interval calculated from a subsampling analysis with data binned by geological periods and subsampled 10,000 times, each with six organic proportion measurements (i.e., fewest measurements in any bin). Cm Cambrian, O Ordovician, S Silurian, D Devonian, C Carboniferous, P Permian, TR Triassic, J Jurassic, K Cretaceous, Pg Paleogene, N Neogene, Q Quaternary and extant. Source data are provided as a Source Data file

relatively low organic content, except ontogenetically immature ones[37].

As an example of early sponges, *V. delicata* with organic-rich and weakly biomineralized spicules may represent a transitional form eventually leading toward fully biomineralized spicules. By extrapolation, we hypothesize that sponges diverged in the Precambrian but Precambrian sponges may have had weakly biomineralized spicules or entirely organic skeletons. The alternative hypothesis—sponges and hence sponge spicules did not diverge until the Cambrian Period—is difficult to reconcile with the presence of bilaterian animals in the Ediacaran Period as evidenced by the abundance of trace fossils[38,39]. A third hypothesis —fully biomineralized sponge spicules evolved in the Precambrian but their fossil record is entirely missing due to taphonomic biases[1] is inconsistent with current understanding of skeletal preservation; as documented by Cambrian examples,

sponge spicules can be preserved in a variety of facies, including carbonates[40], phosphorites[41], cherts[42], and shales[26]. Although we recognize that the absence of evidence is not evidence of absence, it would be a remarkable failure of the fossil record if a group of biomineralized animals is completely missing from the Precambrian, given that extremely delicate and fragile biomineralic scales[43] and weakly biomineralized skeletons such as *Cloudina*[44] are abundantly preserved in Precambrian rocks. Considering all available evidence, we settle in the compromise hypothesis that sponges did diverge but did not develop fully biomineralized spicules in the Precambrian. Indeed, we prefer the scenario that Precambrian sponges produced axial filaments but not biomineralized spicules, because even fragile biomineralic scales[43] and weakly biomineralized animals[44] would not escape from fossilization, as is also demonstrated by *V. delicata* reported here. A ramification of this hypothesis is that biomineralized spicules evolved multiple times and independently among sponge classes, and axial filaments were repeatedly recruited to catalyze sponge spiculogenesis.

A positive paleontological test of this hypothesis requires the discovery of additional weakly biomineralized and aspiculate sponge fossils from the Cambrian and Precambrian. *Cambrowania ovata*, an early Cambrian fossil with organic structures reminiscent of hexactine-based spicules, has been interpreted as a possible juvenile sponge[45], and may represent another case of aspiculate or weakly biomineralized sponges in the early Cambrian. In addition, a number of Ediacaran fossils have been interpreted as non-mineralized sponge animals, although such interpretations have not been widely accepted[18], partly because of the traditional dogma that early sponges must have had biomineralized spicules. A non-exhaustive list includes *Coronacollina* (which has been reconstructed as a sponge animal with a conical body possessing either organic filaments or possibly biomineralized spicules[46]), as well as *Cucullus*, *Liulingjitaenia*, and *Sinospongia* (which are tubular fossils apparently consisting of organic filaments[47]). Notably, phosphatized microfossils in the Ediacaran Doushantuo Formation contain filamentous microstructures up to several microns in diameter (Fig. 7) that resemble monaxonal filaments and were originally interpreted as cylindrical siliceous sponge spicules[48]. Subsequent analyses have shown that they are organic in composition and quadrangular (rectangular) in cross section[17] (Fig. 7b, d), thus decisively falsifying the original interpretation that they represent cylindrical siliceous spicules[48]. However, this does not mean that these filaments cannot be remnants of Ediacaran sponge animals; indeed, these structures may represent axial filaments of early hexactinellids[17]. Given that it is very odd for microbial filaments or otherwise abiotically formed filaments to have a rectangular cross section, and that a typical axial filament of a modern hexactinellid is characterized by a square cross-section (or rectangular cross-section when obliquely cut; Fig. 7e, f)[49–51], it is possible that these organic filaments may represent Ediacaran precursors of axial filaments before hexactinellids acquired biomineralized spicules in the Cambrian. In any case, the lack of biomineralized spicules may be a real signal among Precambrian sponges. If so, it strips a convenient diagnostic feature that can be easily preserved in the rock record, making it more challenging to explore the Precambrian sponge fossil record because the lack of biomineralized spicules is not sufficient to exclude a grouping with total-group sponges.

*V. delicata* can help us reconstruct the sequence of character acquisitions in early sponge evolution (Fig. 8). Because organic axial filaments are functionally essential for the formation of silicean spicules[31], they are inferred to have predated siliceous spicules. It is thus plausible that the last common ancestor of poriferans and perhaps stem-group siliceans may have had only organic filaments, which originally served a function but were

later independently recruited to facilitate spiculogenesis, ultimately evolving into the axial filaments in siliceans. *V. delicata* may represent an evolutionary grade after this recruitment, possessing large axial filaments and thick organic layers, the latter of which may be homologous to the organic lamellae in silicean spicules and organic sheaths in calcarean spicules. Considering their cylindrical axial filaments, the Hetang sponges likely represent poriferans that are phylogenetically outside crown-group hexactinellids and demosponges. Instead, they may be stem-group hexactinellids or even stem-group siliceans, given that it is uncertain whether siliceous hexactine spicules are a synapomorphy of hexactinellids[8].

This interpretation supports the possibility that early spiculate sponges, including stem-group siliceans and stem-group poriferans, may have had weakly biomineralized spicules or lacked spicules. This possibility may also be true for many stem-group hexactinellids, demosponges, and calcareans. Although calcified sponges—archaeocyaths, stromatoporoids, chaetetids, and spinctozoans—are notably abundant in the Paleozoic[52–54], few of them had spicules and none of them have been reported in the Precambrian, hence limiting their power to resolve questions regarding to early evolution of sponge spiculogenesis. The terminal Ediacaran fossil *Namapoikia* has been interpreted as an encrusting poriferan[55], but more work is needed to confirm that it is a calcified encrusting sponge rather than a microbial structure[18]. In any case, *Namapoikia* is not known to have spicules either. Furthermore, the evolution of biocalcification in *Namapoikia* likely requires a pre-existing organic scaffold[55], consistent with the hypothesis that fully biomineralized spicules were likely preceded by organic substrates (i.e., precursors of axial filaments) that can be repeatedly and independently recruited to serve as biomineralization templates and catalysts.

In summary, putative sponge biomarkers, molecular clock estimates, and the presence of bilaterian animals in the Ediacaran Period indicate that sponges and sponge classes diverged in the Precambrian. However, there are no convincing biomineralized sponge spicules in the Precambrian. Early Cambrian sponge spicules described in this paper, as well as some other early Paleozoic sponge spicules, have relatively large proportions of organic material, indicating that early sponges may have had weakly biomineralized spicules or entirely organic skeletons, which unlike fully biomineralized sponge spicules, were less amenable to fossilization. We hypothesize that, although sponges or sponge classes may have diverged in the Precambrian, they independently evolved fully biomineralized spicules at the Precambrian-Cambrian transition.

This hypothesis offers a new search image for Precambrian sponge body fossils. Perhaps early sponges are not preserved as biomineralized spicules, but as carbonaceous remains. In this regard, it is important not to exclude Ediacaran fossils as sponge animals simply because they lack biomineralized spicules, and it is equally important to revisit Ediacaran sponge-like carbonaceous macrofossils that had been previously disregarded as sponges because of the lack of biomineralized spicules. In the end, molecular clocks, biomarker fossils, and spicular fossils must tell a coherent and consistent story about the early evolution of sponge animals.

## Methods

**Stratigraphic background**. A total of 19 sponge body fossils, currently reposited in the Virginia Polytechnic Institute Geosciences Museum (VPIGM, Blacksburg, Virginia, USA), were recovered from the stone coal unit of the Hetang Formation at the Xiaoxi section (29°52.541′N, 118°03.626′E) in the Lantian area (Supplementary Fig. 1). A full description of the geological and stratigraphic background and age constraints of the Hetang Formation in the Lantian area, Anhui Province, South China, was published in ref. [26] The Hetang Formation conformably overlies siliceous rock of the terminal Ediacaran Piyuancun Formation and underlies the

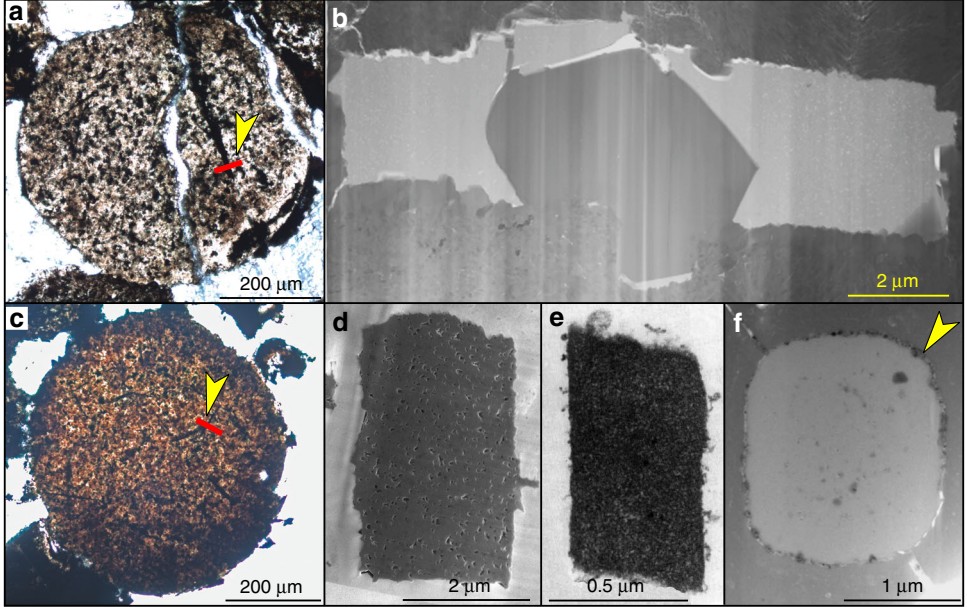

**Fig. 7** Organic filamentous microstructures preserved in Ediacaran microfossils and axial filaments of extant hexactinellid spicules. **a**, **c** Transmitted-light microscopy photographs of phosphatized spherical microfossils in petrographic thin-section obtained from the Ediacaran Doushantuo Formation in South China. Modified from ref. [17]. Arrowheads point to the locations where transverse cross sections (red lines) were prepared across the filamentous microstructures using focus ion beam electron microscopy (FIB-EM). **b** Bright-field scanning/transmission electron microscopy (TEM) photograph of a microstructure cross-section prepared from **a** as an ultra-thin foil, showing that the filament is generally rectangular in cross section. The darker polygon is a euhedral apatite crystal that diagenetically intrudes the filament. **d** SE-SEM micrograph of microstructure cross-section prepared from **c**, showing a rectangular cross section. **e** TEM image of the rectangular cross-section of an axial filament in a spicule of the hexactinellid sponge *Schaudinnia arctica*. Modified from ref. [50]. with permission from author. The slightly lozenge shape of the cross-section is probably due to an oblique section. **f** SEM micrograph of the square cross section (arrowhead) of an axial filament in a hexactinellid sponge spicule. Modified from ref. [51]. with permission from publisher and author.

early Cambrian limestone of the Dachenling Formation (Supplementary Fig. 1). Regionally, the Hetang Formation consists of four lithostratigraphic units, in ascending order, (1) a ~68-m thick mudstone unit rich in phosphorite at the base; (2) a ~30-m thick interval of stone coal (combustible organic-rich mudstone); (3) a ~110-m thick shale and mudstone unit; and (4) a ~110-m thick unit of shale with carbonate nodules[28]. Abundant articulated sponge fossils have been recovered from the stone coal unit which was deposited in a largely ferruginous basinal environment[26,29]. Regional litho- and biostratigraphic correlation on the basis of small shelly fossils, acritarchs, and trilobites in the Hetang Formation indicate the fossiliferous stone coal unit is lower Cambrian Stage 2-3, or ~529−515 Ma (see ref. [26,45]. for more details).

**Optical and electron microscopic analyses**. Specimens were initially examined and photographed on an Olympus SZX7 stereomicrscope connected with an Infinity 1 camera. Well-preserved specimens were then coated with a ~20-nm conductive gold-palladium layer and analyzed using an array of electron microscopic instruments in the Virginia Tech Institute of Critical Technology and Applied Science Nanoscale Characterization and Fabrication Laboratory (VT-ICTAS-NCFL). Secondary electron and backscattered electron scanning electron microscopy (SE-SEM and BSE-SEM), energy dispersive X-ray spectroscopy (EDS), and EDS element mapping were conducted on a FEI Quanta 600FEG environmental SEM, with a pole piece backscattered electron (BSE) solid-state detector (SSD), a secondary electron (SE) Everhart-Thornley detector (ETD), and a Bruker EDX with a silicon drifted detector[56]. The operating voltage in BSE-SEM, SE-SEM, and EDS modes was 5−20 kV in high-vacuum condition. Selected Hetang spicules were subsequently FIB-sectioned on a FEI Helios 600 NanoLab focused ion bean electron microscope (FIB-EM) equipped with a gallium ion beam column for controlled excavation and a high-resolution Elstar Schottky FEG for SEM[57].

**Measurements of organic proportion in sponge spicules**. The distance from the presumed osculum to the aboral end of sponge body fossils was measured and reported as sponge body fossil length; for specimens without a distinguishable osculum, the maximum dimension of sponge body fossils was measured. These measurements are reported in Supplementary Data 1. The diameters of sponge spicules ($d_s$) and axial filaments ($d_{af}$) were measured on the tunnels (i.e., external molds of spicules) and inner core, respectively. When preserved, the maximum thickness of the outer lamella of the cylindrical structure was measured as an

approximation of the thickness of the organic layer ($t_{ol}$). The relative thickness of the axial filament as a percentage of the spicule diameter was calculated as ($d_{af}/d_s$)×100%, and the relative thickness of the organic cylindrical structure as (($2×t_{ol}+d_{af}$)/$d_s$)×100% (i.e., organic proportion). Measurements reported in Supplementary Data 2 are based on published extant and fossil spicules. The relative proportion of the organic structures as a percentage of the spicule diameter was represented by a ratio: when the axial filaments and/or organic layer are preserved, this ratio was calculated as ($d_{af}+(t_{ol}+t_{os})×2)/d_s$×100% (where $t_{os}$ represents the thickness of the outer sheath; purple and orange symbols in Fig. 6a); when only the axial canal is preserved, this ratio was calculated as ($d_{ac}+(t_{ol}+t_{os})×2)/d_s$×100% (where $d_{ac}$ represents the diameter of the axial canal; gray symbols in Fig. 6a). Because axial canals can be enlarged by mineral dissolution[34], the latter method and the gray symbols in Fig. 6a represent maximum estimates of organic proportion in sponge spicules.

A subsampling analysis was conducted on the organic proportion measurements of sponge spicules to determine whether the observed pattern is an artifact of variable sample sizes in geological periods (Fig. 6b). The data, including those derived from axial filament and axial canal measurements, were first binned by geologic periods plus an additional bin for extant sponges. The average organic proportion and its 95% confidence interval were determined for each time bin using a subsampling procedure for standardizing sample sizes. For each time bin, 10,000 subsamples were randomly drawn. In each case, six organic proportion measurements—equal in number to the counts of the bins with the fewest measurements (i.e., Silurian and Triassic)—were sampled without replacement. Then, the average organic proportion of each subsample was calculated, producing a distribution of values. Finally, the 95% confidence interval was determined from the 2.5 and 97.5 percentiles of the distribution. Overall, this procedure tests the null hypothesis that the sponge spicules from different geological periods do not significantly differ with respect to organic proportion. If the 95% confidence intervals of two time bins do not overlap, then there is statistically significant evidence that their spicules have different organic proportions.

**Nomenclatural acts**. This article is published in an electronic journal with an ISSN (2041-1723), and has been archived in PubMed Central. Taxonomic nomenclature published in this article conforms to the requirements of the amended International Code of Zoological Nomenclature (ICZN), and hence is available under ICZN. This publication and the nomenclatural acts it contains have been registered in ZooBank (www.zoobank.org), the proposed online registration system for the

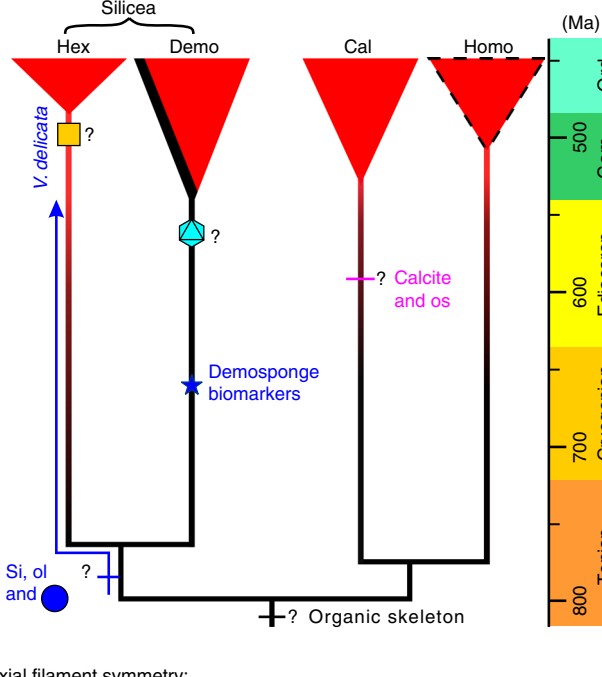

**Fig. 8** Phylogenetic interpretations of *Vasispongia delicata*. The phylogenetic tree is simplified and time-calibrated using molecular clock estimates[13]. It omits eumetazoans and ctenophores so that it stands regardless the monophyly[2–4] vs. paraphyly[1] of the poriferans and the phylogenetic placement of the ctenophores[2,3,58,59]. Although a few molecular phylogenetic analyses give spurious support for the paraphyly of the Silicea[19], most other analyses give decisive support for the monophyly of the Silicea[2–4] (a topology adopted here). A cylindrical axial filament is indicated as a plesiomorphy of crown-group siliceans, because it is also present in the Paleozoic sponges *Cyathophycus loydelli*[24] (possibly a total-group demosponge) and *Lenica*[23] (possibly a stem-group silicean or a stem-group sponge). It is alternatively possible that a cylindrical axial filament could be an autapomorphy of these sponges. This uncertainty, however, does not affect the main conclusion that early sponges may have had weakly biomineralized sponge spicules, which is inferred from the presence of a significant amount of organic matter in many Paleozoic spicules and juvenile spicules of extant sponges[37]. Blue star denotes the possible age of the demosponge biomarker fossil[5,60]. Crown-group classes are denoted by red triangles and their earliest fossil representatives are based on ref. [8]. Dashed line around red triangle indicates the lack of crown-group homoscleromorph fossils in the early Paleozoic. Question marks denote uncertain age constraint or phylogenetic placement of characters. ol organic layer, Si biosilica lamella, os organic sheath, Hex Hexactinellida, Demo Demospongiae, Cal Calcarea, Homo Homoscleromorpha, Cam Cambrian, Ord Ordovician

International Code of Zoological Nomenclature. The ZooBank LSIDs (Life Science Identifiers) can be resolved and the associated information viewed through any standard web browser by appending the LSID to the prefix "http://zoobank.org/". The ZooBank LSID (Life Science Identifier) for this publication is urn:lsid:zoobank.org:pub:51D2A8C0-8440-47CA-B39C-95975CD26AC0. LSIDs for the new genus and species nomenclature are urn:lsid:zoobank.org:act:C2CC4596-0864-432A-9E70-72CF2BB68B5E and urn:lsid:zoobank.org:act:2FAFC2E7-5EBB-4D00-862D-B88E315822CA, respectively.

**Reporting summary**. Further information on research design is available in the Nature Research Reporting Summary linked to this article.

## Data availability

The authors declare that data supporting the findings of this study, including all measurement data, are available within the paper and its supplementary information files. The source data underlying Fig. 6 and Supplementary Fig. 3 are provided as a Source Data file. Supplementary data (https://figshare.com/s/685786bfb660f4840744) have been deposited in the figshare database. Extra data, including all illustrated specimens, are reposited at the Virginia Polytechnic Institute Geosciences Museum (VPIGM) and available from the corresponding author upon request.

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

## Acknowledgements

We thank J. Wang, Dr. S.K. Pandey, and Y. Shao for field assistance, and Dr. H. Tang for assistance in light photography. This research was supported by National Science Foundation (EAR 1528553), State Key Laboratory of Palaeobiology and Stratigraphy of the Nanjing Institute of Geology and Palaeontology (193126), National Natural Science Foundation of China (41130209), Chinese Academy of Sciences (QYZDJ-SSW-DQC009, XDB26000000), and Jiangsu Provincial Department of Science and Technology (BK20161615).

## Author contributions

Q.T. and S.X. designed the research. Q.T., B.W., X.Y. A.D.M. and S.X. conducted fieldwork. Q.T., S.X. and A.D.M. conducted microscopic observation, developed the interpretation, and prepared the manuscript with input from other co-authors.

## Additional information

**Competing interests:** The authors declare no competing interests.

