## [Peer Review File · Nature Communications]

Reviewers' Comments:

Reviewer #1:

Remarks to the Author:

In this paper, Tang and co-authors describe the discovery of biomineralized spicules with large axial filaments and disproportionately large amounts of organic material the Lower Cambrian (though not lowest) Cambrian Hetang Formation from South China. The significance of this material is in what it can tell us about early sponges and possibly Precambrian sponges. One of the big dilemmas in paleontology and biology is the lack of sponge spicules in the Ediacaran. Biomarker and molecular data put sponges back to the Cryogenian and certainly in the Ediacaran. There are well accepted bilaterian fossils and/or trace fossils of bilaterians in the Ediacaran. It is clear that there should be sponges in the Ediacaran. There are a number of purported sponge fossils in the Ediacaran but none have biomineralized spicules preserved. Given the global nature of the Ediacara Biota, it is reasonable to assume that if there were biomineralized spicules not preserved at some site that they would be preserved at another. Yet, there is no record. Tang and authors address this issue with this new discovery. These new fossils do indeed have spicules but they have weakly biomineralized skeletons with much more organic material than younger sponges and that one would predict. This, thus, sets a model for what Ediacaran (or Precambrian) sponges. They may have had axial filaments as suggested by the authors or even been entirely made of OM which is consistent with the lack of spicules in the record and potentially some of the previously described fossils.

This is significant for several reasons. Although debated, many workers suggest that early sponges were spiculate and that the Last common ancestor was speculate. We need to rethink our idea of the Last common ancestor of sponges (this has been true for some time but this discovery adds fuel to that). This discovery constrains when and how the acquisition of spicules happened and also is consistent with the idea that the occurrence of spicules was independent in different groups of sponges.

Sponges are such an integral part of our understanding of the unfolding of animal life on Earth and yet, they are the hardest to constrain in terms of their early evolution. For years, this issue has just gotten more complex without any breakthroughs. This paper is a huge step-forward in thinking and re-thinking about early sponge evolution. It will be of interest to a very wide range of readers. I fully support publication of this paper. It will be a very welcome paper (although I am sure that some will still argue about this) and a very significant and well cited paper.

Minors comments are below.

Line 22 – add “at least some”

The main text should include number of specimens – how common is this? The authors necessarily go right into results but a sentence or two about the material in the main text would be a very helpful addition.

I think that there will be those who find that this is a diagenetic artifact. I think that in Line 104 the authors should say why (for non- sponge workers and for those who will worry about diagenesis especially given other diagenetic features) that they make this interpretation.

The biomarker record has been a point of contention – although organic geochemists have accepted the Love et al., data, others have not fully embraced it. The new work out of that lab has just come out and the reference should be included. This new work really cements the sponge origin of the biomarkers and supports the importance of the Tang et al., paper.

Zumberge, J. Alex, et al. "Demosponge steroid biomarker 26-methylstigmastane provides evidence for Neoproterozoic animals." *Nature ecology & evolution* 2.11 (2018): 1709-1714.

Mary Droser

Reviewer #2:

Remarks to the Author:

This is all conjecture based on some very badly preserved fossils.

There is no meaningful consideration of the preservation of the fossils and when this is meant to be a paper about how fossils could preserve in deep time that is fatal. It is also fatal how there is no discussion of deep time taphonomies. The late Precambrian oceans were precipitating silica not dissolving it. There is no missing silica in the Precambrian it is hugely abundant, for billions of years from the many famous Chert deposits that preserve microbes (e.g. Gunflint) through to the rapidly formed early silica cements that helped preserve soft bodied fossils in siliciclastic rocks (e.g. Ediacara), silica is absolutely everywhere.

What also is fatal is the central logic of the paper which runs thus: these fossil sponges are badly preserved therefore maybe all early fossil sponges are badly preserved therefore sponges could be much older and just missing from the record. Maybe these fossils are just bad. Mainly because they are preserved as compression fossils not silica or phosphate replication. They are not comparing like taphonomies. It doesn't help the argument that there are many excellently preserved sponges from the early Cambrian interval that is the subject of the manuscript discussing that don't show any of the characters that can only be found in these badly preserved specimens. It is also worth noting that you still found these fossils, they may be badly preserved but they are still there. So why does this imply that earlier fossils are not preservable? You can't find fossils and then make the case that such fossils can't be found.

The whole premise of the paper is based on the completely false statements found in Lines 17-19. This completely misrepresents and distorts the state of the debate around the origin of animals and sponges in particular. The debate is not why there are no Precambrian spicules as presented here but whether the fossil record is good or if the molecular record is good. The field as a whole is not simply seeking to explain the discord by understanding why the paleontological record is bad on the assumption that the molecular record is good. Therefore, the third hypothesis excluded here is that the molecular record is garbage and provides no good evidence for the early evolution of animals. This is a mainstream view held by countless palaeontologists and cannot simply be ignored by the present authors for convenience. As a result, the manuscript needs to be completely rewritten such that it fairly represents the state of play in the field, and to consider more broadly the alternative hypotheses which so readily come to mind.

Figures are not clear and need a lot of work. There need to be drawings of the specimens and much more space given to description of the material. Then there needs to be some really geology. Thin section work. Mineralogy. Fabric mapping. Something about metamorphism/metasomatism/post depositional history at the site. To go forwards the authors should focus on the taphonomy of these fossils. Then you need to consider in detail why the taphonomy of these fossils may explain why they appear different to coeval fossils from different taphonomies.

Reviewer #3:

Remarks to the Author:

This submission argues that the reported early Cambrian Hetang sponges, like some other early Paleozoic sponges, had weakly biomineralized spicules with large axial filaments and disproportionately large amounts of organic material. Thus, the authors state that early sponges had weakly biomineralized spicules, and this may explain the absence of sponge material in the Precambrian fossil record, even when molecular biomarkers suggest their presence.

The general hypothesis that early sponges lacked mineralised spicules is appealing, and indeed has been in general currency for a while. I am certainly supportive of this idea – but on balance I do not feel that this contribution markedly strengthens or contributes to the hypothesis.

First, we already have evidence of diverse sponges with substantial organic components to their spicules in the Cambrian and Ordovician, as noted by the authors. The only novelty presented here is that the Hetang sponges show far larger diameters of axial filaments and organic layers, and a markedly wider range, than all others documented – even compared to all the Cambrian (and older than the Hetang) records compiled. So does this really constitute a trend?? If so, this needs to be demonstrated statistically.

I am also rather concerned with the oft repeated phrase 'Hetang sponges/spicules'. Does this refer to single taxon, or multiple taxa? If only one taxon, why could this not simply be the character of a single species, rather than being representative of the state of 'early sponges'? Extrapolating the evolutionary trajectory of a whole phylum from one taxon would be a massive stretch!

Secondly, an issue with current early sponge phylogenies is that they are based only on spiculate taxa. Non-spiculate calcified sponges, such as all archaeocyaths, and most ancient stromatoporoids, chaetetids and spinctozoans (which are probably convergent 'grades' of both calcified demosponges and calcareans) are largely ignored because of the difficulty of placement within the Porifera - but few would argue that they are not sponges. Archaeocyaths almost certainly biomineralised via calcification of a pre-existing organic scaffold, and are a dominant and species-rich part of the early Cambrian biota. As the authors' state '... the lack of biomineralized spicules is not sufficient to exclude a grouping with total-group sponges.' So why are they ignored here?

Likewise, what about the Ediacaran taxon Namapoikia? This has been proposed to be an aspiculate, sponge with a calcified basal skeleton also formed from a pre-existing organic scaffold. Discussion of this taxon is therefore highly pertinent to the arguments of Tang et al., and should be used either to support their assertion, or they may wish to dismiss a poriferan affinity for this taxon.

In sum, as spicules may have evolved independently among sponge classes, the conclusion of Tang et al., that '...early spiculate sponges, including stem-group siliceans and stem-group poriferans, may have weakly biomineralized spicules or even entirely organic skeletons' might well be true. But this submission does not offer either definitive new data or consider important contextual studies that have contributed to this debate.

Reviewers' comments:

Reviewer #1 (Remarks to the Author):

In this paper, Tang and co-authors describe the discovery of biomineralized spicules with large axial filaments and disproportionately large amounts of organic material the Lower Cambrian (though not lowest) Cambrian Hetang Formation from South China. The significance of this material is in what it can tell us about early sponges and possibly Precambrian sponges. One of the big dilemmas in paleontology and biology is the lack of sponge spicules in the Ediacaran. Biomarker and molecular data put sponges back to the Cryogenian and certainly in the Ediacaran. There are well accepted bilaterian fossils and/or trace fossils of bilaterians in the Ediacaran. It is clear that there should be sponges in the Ediacaran. There are a number of purported sponge fossils in the Ediacaran but none have biomineralized spicules preserved. Given the global nature of the Ediacara Biota, it is reasonable to assume that if there were biomineralized spicules not preserved at some site that they would be preserved at another. Yet, there is no record. Tang and authors address this issue with this new discovery. These new fossils do indeed have spicules but they have weakly biomineralized skeletons with much more organic material than younger sponges and that one would predict. This, thus, sets a model for what Ediacaran (or Precambrian) sponges. They may have had axial filaments as suggested by the authors or even been entirely made of OM which is consistent with the lack of spicules in the record and potentially some of the previously described fossils.

This is significant for several reasons. Although debated, many workers suggest that early sponges were spiculate and that the Last common ancestor was spiculate. We need to rethink our idea of the Last common ancestor of sponges (this has been true for some time but this discovery adds fuel to that). This discovery constrains when and how the acquisition of spicules happened and also is consistent with the idea that the occurrence of spicules was independent in different groups of sponges.

Sponges are such an integral part of our understanding of the unfolding of animal life on Earth and yet, they are the hardest to constrain in terms of their early evolution. For years, this issue has just gotten more complex without any breakthroughs. This paper is a huge step-forward in thinking and re-thinking about early sponge evolution. It will be of interest to a very wide range of readers. I fully support publication of this paper. It will be a very welcome paper (although I am sure that some will still argue about this) and a very significant and well cited paper.

Minors comments are below.

Line 22 – add “at least some”

The sentence has been removed.

The main text should include number of specimens – how common is this? The authors necessarily go right into results but a sentence or two about the material in the main text would be a very helpful addition.

The number of specimens has been added in the revised main text.

Revised text: In total, 19 well-preserved specimens of *V. delicata* were collected from organic-rich mudstones of the lower Hetang Formation. (Line 57–58)

I think that there will be those who find that this is a diagenetic artifact. I think that in Line 104 the authors should say why (for non- sponge workers and for those who will worry about diagenesis especially given other diagenetic features) that they make this interpretation.

This interpretation has been explained in the revised text. Specifically, a section on "Preservation" has been added to the revised text to explain why some structures are interpreted as taphonomic artifacts whereas others are regarded as biological features.

Revised text: More importantly, the cylindrical structure in the center of the spicules is of biological origin. This interpretation is supported by the observation that the cylindrical structure is consistently composed of an inner core surrounded by an outer lamella (Fig. 2a–c and 3a–c, f and 4a) or outer siliceous layer (Fig. 4c), and that the inner core is ornamented with exquisitely preserved ridges and tubercles (Fig. 1e–g and supplementary Fig. 4). (Line 132–136)

The biomarker record has been a point of contention – although organic geochemists have accepted the Love et al., data, others have not fully embraced it. The new work out of that lab has just come out and the reference should be included. This new work really cements the sponge origin of the biomarkers and supports the importance of the Tang et al., paper.

Zumberge, J. Alex, et al. "Demosponge steroid biomarker 26-methylstigmastane provides evidence for Neoproterozoic animals." *Nature ecology & evolution* 2.11 (2018): 1709-1714.

The paper has been cited in the revised text.

Mary Droser

Reviewer #2 (Remarks to the Author):

This is all conjecture based on some very badly preserved fossils.

We regretfully disagree that the Hetang sponges are “badly preserved fossils”. Instead, those are exceptionally preserved sponges in the sense that they show well-preserved axial filament, which are almost always missing in Phanerozoic sponge fossils. And it is the axial filaments that allow us to draw our conclusions.

A section on "Preservation" has been added to the revised text, explaining why the new fossils are considered exceptionally preserved. This section ends with the following sentence: Considering that the Cambrian sponge fossil record is dominated by disarticulated spicules with few microstructures, *V. delicata* is considered an exceptionally preserved body fossil with exquisitely preserved organic component of the spicules. (Line 136–139)

There is no meaningful consideration of the preservation of the fossils and when this is meant to

be a paper about how fossils could preserve in deep time that is fatal. It is also fatal how there is no discussion of deep time taphonomies. The late Precambrian oceans were precipitating silica not dissolving it. There is no missing silica in the Precambrian it is hugely abundant, for billions of years from the many famous Chert deposits that preserve microbes (e.g. Gunflint) through to the rapidly formed early silica cements that helped preserve soft bodied fossils in siliciclastic rocks (e.g. Ediacara), silica is absolutely everywhere.

We agree that the Precambrian ocean was probably full of silica. However, this has nothing to do with our argument that Precambrian sponges did not develop siliceous spicules, instead they may develop weakly mineralized or even fully organic skeletons. Simply because Precambrian ocean had abundant silica does not mean that Precambrian sponges should have had siliceous skeletons. In addition, the demineralization of some Hetang spicules did not occur in silica-rich Precambrian oceans or in Cambrian oceans. Rather, such demineralization is a taphonomic phenomenon that occurred long after fossilization.

What also is fatal is the central logic of the paper which runs thus: these fossil sponges are badly preserved therefore maybe all early fossil sponges are badly preserved therefore sponges could be much older and just missing from the record. Maybe these fossils are just bad. Mainly because they are preserved as compression fossils not silica or phosphate replication. They are not comparing like taphonomies. It doesn't help the argument that there are many excellently preserved sponges from the early Cambrian interval that is the subject of the manuscript discussing that don't show any of the characters that can only be found in these badly preserved specimens. It is also worth noting that you still found these fossils, they may be badly preserved but they are still there. So why does this imply that earlier fossils are not preservable? You can't find fossils and then make the case that such fossils can't be found.

Please see our response below.

The whole premise of the paper is based on the completely false statements found in Lines 17-19. This completely misrepresents and distorts the state of the debate around the origin of animals and sponges in particular. The debate is not why there are no Precambrian spicules as presented here but whether the fossil record is good or if the molecular record is good. The field as a whole is not simply seeking to explain the discord by understanding why the paleontological record is bad on the assumption that the molecular record is good. Therefore, the third hypothesis excluded here is that the molecular record is garbage and provides no good evidence for the early evolution of animals. This is a mainstream view held by countless palaeontologists and cannot simply be ignored by the present authors for convenience. As a result, the manuscript needs to be completely rewritten such that it fairly represents the state of play in the field, and to consider more broadly the alternative hypotheses which so readily come to mind.

We do not agree that "molecular record is garbage and provides no good evidence for the early evolution of animals", nor that "This is a mainstream view held by countless palaeontologists". We would like to point out that the statement in Lines 17-18 ("Therefore, sponges either evolved spiculogenesis long after their origin or their spicules were not amenable to fossilization in the Precambrian.") was based on Sperling et al. 2010, which has guided the field in the last decade, and our statement is not a distortion of the field as claimed by the reviewer. The reviewer's bias towards molecular clock and biomarker data is based on his or her unjustified dismissal of a

number of recent molecular clock studies (Sperling et al., 2010; Erwin et al., 2011; Dos Reis et al., 2015; Dohrmann and Wörheide, 2017; Schuster et al., 2018, and many others) and biomarker studies (Love et al., 2009; Gold et al., 2016; Zumberge et al., 2018). It is important to point out that many of these recent work on molecular clocks and biomarkers are led or co-authored by respected paleontologists. To the best of our knowledge, no paleontologists have published any serious rebuttal of the cited molecular clock studies (we understand that does not mean all paleontologists agree with the molecular clocks), and only a single paleontologist (Antcliffe, 2013) has questioned the biomarker data by Love et al. (2009), but the recent work by Zumberge et al. (2018) has vindicated the Love et al. 2009 study. So we think that reviewer's bias is unjustified and is at best a minority view, certainly not a "mainstream view held by countless palaeontologists". Additionally, as pointed out by reviewer 1, the presence of unambiguous bilaterians in the Ediacaran Period also demands that sponges diverged in the Precambrian if sponges are monophyletic, and that siliceans diverged in the Precambrian if sponges are paraphyletic.

With this "premise", we would like to respond to the reviewer's previous comments about the logic of our argument. Certainly, we do not agree with reviewer's summary about our logic. Our inference that "sponges could be much older" than Cambrian is based on molecular clock and biomarker data. But biomineralized spicules are missing from the Precambrian, leaving a significant discrepancy between the molecular/biomarker data and paleontological record. As we do not agree with the reviewer's dismissal of the molecular clock estimates and biomarker data, this discrepancy means that either Precambrian spicules are somehow not preserved or Precambrian sponges did not have biomineralized spicules. The first hypothesis has been discussed extensively in Sperling et al. (2010), and our manuscript addresses the second hypothesis, which predicts independent origins of biomineralized spicules among sponge classes and the presence of transitional weakly biomineralized spicules among early sponges. The well-preserved *Hetang* sponges and many other early Paleozoic sponges are interpreted as such expected transitional forms, as their spicules contain a large proportion of organic materials and represent weakly biomineralized structures. With this trend (as shown in Fig. 6), it is likely that those Precambrian sponges may also only precipitate weakly mineralized or even fully organic skeletons. We propose that several putative sponge fossils from the Ediacaran can further test this hypothesis. In the revised manuscript, we highlight phosphatized microfossils with filamentous structures from the Ediacaran Doushantuo Formation. These filaments are organic in composition and have a rectangular cross-section (Fig. 7 in the revised manuscript), and are interpreted as Ediacaran precursors of axial filaments that were subsequently recruited to catalyze spiculogenesis in hexactinellid spicules during the Cambrian. In the revised manuscript, we outline future work to further test this hypothesis and offer a new search image for Precambrian sponge fossils, pointing out that Precambrian sponges may lack biomineralized spicules and may be preserved as carbonaceous compressions.

Figures are not clear and need a lot of work. There need to be drawings of the specimens and much more space given to description of the material. Then there needs to be some really geology. Thin section work. Mineralogy. Fabric mapping. Something about metamorphism/metasomatism/post depositional history at the site. To go forwards the authors

should focus on the taphonomy of these fossils. Then you need to consider in detail why the taphonomy of these fossils may explain why they appear different to coeval fossils from different taphonomies.

We have added a new section on fossil preservation. In this section, we have explained why some features are interpreted as taphonomic artifacts whereas others as biological features. Regional geology and stratigraphy, paleoredox conditions, and mineralogy of the fossiliferous unit have been previously published and relevant literatures are cited in the manuscript. The Hetang Formation is little metamorphosed. It is known for exceptional preservation of soft-bodied animals and articulated sponge body fossils. Anoxic depositional environments (as evidenced by Fe speciation data, redox-sensitive trace element concentrations, and framboidal pyrite size distributions) as well as fine-grained clay minerals may have partially contributed to the exceptional fossil preservation in this unit.

Reviewer #3 (Remarks to the Author):

This submission argues that the reported early Cambrian Hetang sponges, like some other early Paleozoic sponges, had weakly biomineralized spicules with large axial filaments and disproportionately large amounts of organic material. Thus, the authors state that early sponges had weakly biomineralized spicules, and this may explain the absence of sponge material in the Precambrian fossil record, even when molecular biomarkers suggest their presence.

The general hypothesis that early sponges lacked mineralised spicules is appealing, and indeed has been in general currency for a while. I am certainly supportive of this idea – but on balance I do not feel that this contribution markedly strengthens or contributes to the hypothesis.

First, we already have evidence of diverse sponges with substantial organic components to their spicules in the Cambrian and Ordovician, as noted by the authors. The only novelty presented here is that the Hetang sponges show far larger diameters of axial filaments and organic layers, and a markedly wider range, than all others documented – even compared to all the Cambrian (and older than the Hetang) records compiled. So does this really constitute a trend?? If so, this needs to be demonstrated statistically.

We agree with the reviewer that the general hypothesis presented in the manuscript is appealing and has been implicitly stated in several publications. Our manuscript formalize this hypothesis, provides quantitative data to explicitly test this hypothesis, outlines ways to further test the ramifications of this hypothesis, and offers a new search image for Precambrian sponge fossils. The sponge fossils from the Hetang Formation contain well-preserved and exceptionally large axial filaments that directly address the problem of expected transitional forms between aspiculate and fully spiculate siliceans. This is one of the major highlights of the manuscript.

The large range of the organic proportion in Hetang spicules is partly due to degradation (including devolatilization and oxidation), which has been clarified in the revised text.

With regard to the trend, we have provided statistical analysis to show that there is indeed a trend that Cambrian spicules have greater organic content than younger ones (Fig. 6b). We emphasize

that because axial filaments are rarely preserved in the fossil record (particularly post-Cambrian spicules), and as such we had to depend on the preservation of axial canal as a proxy for axial filament. This approach tend to overestimate the size of post-Cambrian axial filaments, but despite this Cambrian axial filaments are still statistically larger than post-Cambrian ones (Fig. 6b).

Revised text: The variation in axial filament diameter and organic layer thickness may be partly due to taphonomic degradation; given that thermal maturation, devolatilization, and chemical oxidation of organic structures tend to result in volume reduction, the upper end of the range of measurements is more likely to approach the original sizes. (Line 183–186)

I am also rather concerned with the oft repeated phrase ‘Hetang sponges/spicules’. Does this refer to single taxon, or multiple taxa? If only one taxon, why could this not simply be the character of a single species, rather than being representative of the state of ‘early sponges’? Extrapolating the evolutionary trajectory of a whole phylum from one taxon would be a massive stretch!

This is a valid concern. In the revised manuscript, we provide systematic treatment of the Hetang sponge fossils. They indeed belong to a single species. We acknowledge that a single species is not sufficient to rewrite the evolutionary history of sponge spiculogenesis, but we also recognize that we have to start from somewhere. To partially alleviate this problem, we assembled a compilation of sponge spicules and identify a statistically significant trend (Fig. 6) to substantiate our hypothesis. We present a candidate for precursor axial filament from the Ediacaran Doushantuo Formation (Fig. 7). We also propose future work (e.g., the discovery of additional early sponge fossils with weakly biomineralized spicules and the identification of additional aspiculate sponge fossils from the Precambrian) to further test this hypothesis. In the end, the fossil record and the molecular record have to tell a consistent story. At the present, we feel that there is insufficient evidence for us to dump the molecular clock estimates as “garbage” and to explain the lack of Precambrian spicules (missing glass) as a massive taphonomic failure. More likely, the solution will come from a new search image of Precambrian sponges.

Secondly, an issue with current early sponge phylogenies is that they are based only on spiculate taxa. Non-spiculate calcified sponges, such as all archaeocyaths, and most ancient stromatoporoids, chaetetids and spinctozoans (which are probably convergent ‘grades’ of both calcified demosponges and calcareans) are largely ignored because of the difficulty of placement within the Porifera - but few would argue that they are not sponges. Archaeocyaths almost certainly biomineralised via calcification of a pre-existing organic scaffold, and are a dominant and species-rich part of the early Cambrian biota. As the authors’ state ‘... the lack of biomineralized spicules is not sufficient to exclude a grouping with total-group sponges.’ So why are they ignored here?

Likewise, what about the Ediacaran taxon Namapoikia? This has been proposed to be an aspiculate, sponge with a calcified basal skeleton also formed from a pre-existing organic scaffold. Discussion of this taxon is therefore highly pertinent to the arguments of Tang et al., and should be used either to support their assertion, or they may wish to dismiss a poriferan affinity for this taxon.

We agree that archaeocyaths, stromatoporoids, chaetetids, and spinctozoans are likely sponges, although their exact phylogenetic positions within the Porifera are uncertain and they do not occur

in the Precambrian. As the reviewer points out, the possible sponge fossil *Namapoikia* from the terminal Ediacaran Period indicates the pre-existence of an organic scaffold, consistent with our hypothesis of organic precursors for both spiculogenesis and massive biocalcification. In the revised manuscript, we have added a paragraph to discuss these biocalcified fossils.

Revised text: Although aspiculate calcified sponges— archeocyaths, stromatoporoids, chaetetids, and spinctozoans—are notably abundant in the Paleozoic³⁷⁻³⁹, few have been reported in the Precambrian and their exact phylogenetic positions within the sponges are uncertain, hence limiting their power to resolve questions regarding to early sponge biomineralization. The terminal Ediacaran fossil *Namapoikia* has been interpreted as an encrusting poriferan⁴⁰, but more work is needed to confirm that it is a calcified encrusting sponge rather than a microbial structure¹². In any case, most of these calcifying sponge fossils do not have spicules, and the evolution of massive biocalcification requires a pre-existing organic scaffold⁴⁰, consistent with the hypothesis that fully biomineralized spicules did not originate in the last common ancestor of sponges, but rather, they independently evolved later in several sponge classes, repeatedly recruiting the same organic substrates as biomineralization templates and catalysts. (Line 251–262)

In sum, as spicules may have evolved independently among sponge classes, the conclusion of Tang et al., that ‘...early spiculate sponges, including stem-group siliceans and stem-group poriferans, may have weakly biomineralized spicules or even entirely organic skeletons’ might well be true. But this submission does not offer either definitive new data or consider important contextual studies that have contributed to this debate.

We feel that the preservation of large axial filaments is something new and directly relevant to the current debate. Axial filaments are extremely rare in fossil record (see Fig. 6 for the rather few examples of axial filament fossils in our compilation), and our report of well-preserved axial filament in the organic-rich spicules from the early Cambrian Hetang Formation is a significant discovery. The Hetang fossils represent a step forward to fill the gap between the aspiculate and fully spiculate siliceans. Additionally, we offer a new search image for Precambrian sponge fossils and present a possible candidate of precursor axial filaments from the Ediacaran Period (Fig. 7).

References

- Antcliffe, J. B., 2013, Questioning the evidence of organic compounds called sponge biomarkers: *Palaeontology*, v. 56, no. 5, p. 917–925.
- Dohrmann, M., and Wörheide, G., 2017, Dating early animal evolution using phylogenomic data: *Scientific Reports*, v. 7, no. 1, p. 3599.
- Dos Reis, M., Thawornwattana, Y., Angelis, K., Telford, Maximilian J., Donoghue, Philip C. J., and Yang, Z., 2015, Uncertainty in the Timing of Origin of Animals and the Limits of Precision in Molecular Timescales: *Current Biology*, v. 25, no. 22, p. 2939–2950.
- Erwin, D. H., Laflamme, M., Tweedt, S. M., Sperling, E. A., Pisani, D., and Peterson, K. J., 2011, The Cambrian conundrum: Early divergence and later ecological success in the early history of animals: *Science*, v. 334, p. 1091–1097.
- Gold, D. A., Grabenstatter, J., de Mendoza, A., Riesgo, A., Ruiz-Trillo, I., and Summons, R. E., 2016, Sterol and genomic analyses validate the sponge biomarker hypothesis:

- Proceedings of the National Academy of Sciences of the United States of America, v. 113, no. 10, p. 2684–2689.
- Love, G. D., Grosjean, E., Stalvies, C., Fike, D. A., Grotzinger, J. P., Bradley, A. S., Kelly, A. E., Bhatia, M., Meredith, W., Snape, C. E., Bowring, S. A., Condon, D. J., and Summons, R. E., 2009, Fossil steroids record the appearance of Demospongiae during the Cryogenian period: *Nature*, v. 457, p. 718–721.
- Schuster, A., Vargas, S., Knapp, I. S., Pomponi, S. A., Toonen, R. J., Erpenbeck, D., and Wörheide, G., 2018, Divergence times in demosponges (Porifera): first insights from new mitogenomes and the inclusion of fossils in a birth-death clock model: *BMC Evolutionary Biology*, v. 18, no. 1, p. 114.
- Sperling, E. A., Robinson, J. M., Pisani, D., and Peterson, K. J., 2010, Where's the glass? Biomarkers, molecular clocks, and microRNAs suggest a 200-Myr missing Precambrian fossil record of siliceous sponge spicules: *Geobiology*, v. 8, p. 24–36.
- Zumberge, J. A., Love, G. D., Cárdenas, P., Sperling, E. A., Gunasekera, S., Rohrsen, M., Grosjean, E., Grotzinger, J. P., and Summons, R. E., 2018, Demosponge steroid biomarker 26-methylstigmastane provides evidence for Neoproterozoic animals: *Nature Ecology & Evolution*, v. 2, no. 11, p. 1709–1714.

Reviewers' Comments:

Reviewer #1:

Remarks to the Author:

I had relatively minor comments for the first review. These comments have all been addressed. I am very happy with the new revised version of the paper. I have also read through the first reviews and response to reviews by the authors.

I found review #2 inconsistent with the data (both before and after the review) and also that the reviewer was biased against the biomarker record and even the potential of sponges in the Precambrian. I think that the authors have dealt with this review as well as could be done.

I think that reviewer # 3 had some very helpful comments and edits and I think that authors have addressed these adequately.

The early record of sponges is hotly debated. There are those who will never accept that there are sponges even in the Ediacaran in spite of evidence to the contrary and in spite of the record of bilaterians in the Ediacaran. What is more exciting is to try and constrain this record. The discovery of these fossils is hugely important to the understanding of evolution of early life. This is exactly the kind of discovery that will help truly begin to constrain our record. I find the fossils compelling as I did with the original paper.

I fully support publication of this paper in a timely manner!

Reviewer #3:

Remarks to the Author:

I have read the response to reviews, and the revised MS.

First, a significant new paper is now published (Nettersheim et al., 2019, Nature Ecology and Evolution) which raises considerable doubt that the fossil lipids 24-isopropylcholestane and 26-methylstigmastane are in fact diagnostic for demosponges. Nettersheim et al. show that the biosynthesis of 24-isopropylcholestane and 26-methylstigmastane precursors is common among early-branching unicellular Rhizaria – that is, heterotrophic protists. Negating these lipids as sponge biomarkers may therefore place the oldest evidence for animals closer to the Cambrian Explosion, or just before. This is so significant that substantial parts of the MS will need to be revised, and indeed the overall argument will need to be totally re-framed.

I should also add that there are many other papers apart from Antcliffe et al., 2014, such as those of Budd, and Botting, which question whether the fossil lipids 24-isopropylcholestane and 26-methylstigmastane are indeed diagnostic for demosponges.

Figure 6 presents quantitative data to explicitly test the hypothesis they raise. I am, however, a little confused, as the data presented for *V. delicata* is based on axial filament diameter, but on axial canal diameter for everything else. In 6a, we have only one taxon (*V. delicata*) that is very high, but also the Ordovician taxa seem to show a large size compared to the rest of the Phanerozoic. So it is not clear to me how these data translate into the data plotted on 6b. Surely also, the data need to be normalised?

The authors confirm that the Hetang sponges/spicules in fact represents only one species. I therefore

remain concerned that this is simply the character of that single taxon, rather than being representative of the state of 'early sponges'? Hence extrapolation of the evolutionary trajectory of a whole phylum from one taxon is hardly, statistically or otherwise, valid. A new candidate for preservation of a precursor axial filament from the Ediacaran Doushantuo Formation has now been slipped in (Fig. 7) to aid the argument, but this has not received a fulsome analysis.

New text has been added to mention the importance of aspiculate calcified sponges. The fact that their exact phylogenetic positions within the Porifera are uncertain is not really relevant here. The Hetang sponge is also of uncertain phylogenetic position, so the statement that they have 'limitingpower to resolve questions regarding to early sponge biomineralization' is a red herring and erroneous. Why should it be accepted for one taxon (the Hetang sponge) and not many other diverse taxa (indeed whole 'groups' - the archaeocyaths, stromatoporoids, chaetetids and sphinctozoans).

Finally, how novel is this idea, and indeed these data? The organic skeleton precursor hypothesis has a long standing currency for most groups of biomineralizers. We are left then with the finding of the preservation of large axial filaments in a single species of Cambrian sponge. While this does contribute to the debate, I do not feel it is of sufficient importance to warrant publication in Nat. Comms.

Reviewers' comments:

Reviewer #1 (Remarks to the Author):

I had relatively minor comments for the first review. These comments have all been addressed. I am very happy with the new revised version of the paper. I have also read through the first reviews and response to reviews by the authors.

I found review #2 inconsistent with the data (both before and after the review) and also that the reviewer was biased against the biomarker record and even the potential of sponges in the Precambrian. I think that the authors have dealt with this review as well as could be done.

I think that reviewer # 3 had some very helpful comments and edits and I think that authors have addressed these adequately.

The early record of sponges is hotly debated. There are those who will never accept that there are sponges even in the Ediacaran in spite of evidence to the contrary and in spite of the record of bilaterians in the Ediacaran. What is more exciting is to try and constrain this record. The discovery of these fossils is hugely important to the understanding of evolution of early life. This is exactly the kind of discovery that will help truly begin to constrain our record. I find the fossils compelling as I did with the original paper.

I fully support publication of this paper in a timely manner!

Reviewer #3 (Remarks to the Author):

I have read the response to reviews, and the revised MS.

First, a significant new paper is now published (Nettersheim et al., 2019, Nature Ecology and Evolution) which raises considerable doubt that the fossil lipids 24-isopropylcholestane and 26-methylstigmastane are in fact diagnostic for demosponges. Nettersheim et al. show that the biosynthesis of 24-isopropylcholestane and 26-methylstigmastane precursors is common among early-branching unicellular Rhizaria – that is, heterotrophic protists. Negating these lipids as sponge biomarkers may therefore place the oldest evidence for animals closer to the Cambrian Explosion, or just before. This is so significant that substantial parts of the MS will need to be revised, and indeed the overall argument will need to be totally re-framed.

This new paper (Nettersheim et al., 2019) came out after our manuscript was submitted and hence was not cited. It is now cited and briefly discussed in the revised manuscript. Nettersheim et al. (2019) questioned the Cryogenian lipids 24-ipc and 26-mes as sponge biomarkers. Organic geochemists and molecular biologists continue

to work on sponge biomarkers, and it is unlikely that Nettersheim et al., 2019 have the last word on this controversial subject. More importantly, although Nettersheim et al. (2019) provide an alternative interpretation of 24-ipc and 26-mes, they do not falsify the existence of Precambrian sponges. In the revised manuscript, we cite three lines of evidence supporting a Precambrian divergence of sponges. In addition to the biomarker data, molecular clock estimates and Ediacaran bilaterian fossils also support a Precambrian divergence of sponges. Recent molecular clock studies with improved taxonomic sampling of sponges and independent of the Cryogenian biomarkers as a calibration indicate a Neoproterozoic origin of sponges (Dohrmann and Wörheide, 2017; Schuster et al., 2018). Perhaps the most robust evidence for a Precambrian divergence of sponges comes from the paleontological record. Bilaterian animal fossils—as evidenced by trails, tracks, and burrows, as well as body fossils such as *Kimberella*—have been reported in the Ediacaran Period (Xiao and Laflamme, 2009; Darroch et al., 2018; Wood et al., 2019). Given our understanding of animal phylogeny, these bilaterian animal fossils mandate the divergence of sponges or sponge classes (if sponges are paraphyletic) in the Ediacaran Period or earlier. Therefore, the divergence of sponges in the Precambrian is well supported by multiple independent line of evidence. The recent paper by Nettersheim et al. (2019), while important, does not undermine how we frame our manuscript. In the revision, we acknowledge the Nettersheim et al. (2019) paper, but have provided a narrative that includes biomarkers, molecular clocks, and Ediacaran bilaterian fossils to support the Precambrian divergence of sponges.

Revised text: “Sponge animals, either paraphyletic at the base of the animal tree¹ or forming a monophyletic clade that is the sister group of all other animals²⁻⁴, likely diverged in the Precambrian. Biomarker fossils suggest that sponge classes diverged no later than the Cryogenian Period^{5,6}, although their interpretations remain a matter of debate⁷⁻¹⁰. Molecular clock studies¹¹, including recent ones with improved taxonomic sampling and independent of the aforementioned biomarkers as a calibration^{12,13}, point to a similar antiquity of sponge classes. More importantly, the presence of bilaterian animals in the Ediacaran Period¹⁴⁻¹⁶ strongly indicates the divergence of sponges and even sponge classes in the Precambrian, particularly if sponges are paraphyletic¹.” (Line 30–37)

I should also add that there are many other papers apart from Antcliffe et al., 2014, such as those of Budd, and Botting, which question whether the fossil lipids 24-isopropylcholestane and 26-methylstigmastane are indeed diagnostic for demosponges.

These authors (Antcliffe, 2013; Budd, 2013; Botting and Muir, 2018; Nettersheim et al., 2019) have been cited in the revised MS to reflect the controversial nature of the Cryogenian sponge biomarkers.

Revised text: “Biomarker fossils suggest that sponge classes diverged no later than the Cryogenian Period^{5,6}, although their interpretations remain a matter of debate⁷⁻¹⁰.”

(Line 32–33)

Figure 6 presents quantitative data to explicitly test the hypothesis they raise. I am, however, a little confused, as the data presented for *V. delicata* is based on axial filament diameter, but on axial canal diameter for everything else. In 6a, we have only one taxon (*V. delicata*) that is very high, but also the Ordovician taxa seem to show a large size compared to the rest of the Phanerozoic. So it is not clear to me how these data translate into the data plotted on 6b. Surely also, the data need to be normalised? The data presented in Fig. 6b is both standardized and normalized. This has been explained in the “Materials and Methods” section. Briefly, the raw data, including those derived from axial filament and axial canal measurements, were first binned by geologic periods plus an additional bin for extant sponges. The average organic proportion and its 95% confidence interval were determined for each time bin using a subsampling procedure to standardize sample sizes. For each time bin, 10,000 subsamples were randomly drawn. In each case, six organic proportion measurements—equal in number to the counts of the bins with the fewest measurements (i.e., Silurian and Triassic)—were sampled without replacement. Then, the average organic proportion of each subsample was calculated, producing a distribution of values. Finally, the 95% confidence interval was determined from the 2.5 and 97.5 percentiles of the distribution. Overall, this procedure tests the null hypothesis that the sponge spicules from different geological periods do not significantly differ with respect to organic proportion. If the 95% confidence intervals of two time bins do not overlap, then there is statistically significant evidence that their spicules have different organic proportions.

To avoid potential confusion to the readers, we have added a sentence in the figure caption to briefly explain how data plotted in Fig. 6b were generated.

Revised text: “Mean and 95% confidence interval calculated from a subsampling analysis with data binned by geological periods and subsampled 10,000 times, each with six organic proportion measurements (i.e., fewest measurements in any bin).”
(Line 580–583)

The authors confirm that the Hetang sponges/spicules in fact represents only one species. I therefore remain concerned that this is simply the character of that single taxon, rather than being representative of the state of ‘early sponges’? Hence extrapolation of the evolutionary trajectory of a whole phylum from one taxon is hardly, statistically or otherwise, valid. A new candidate for preservation of a precursor axial filament from the Ediacaran Doushantuo Formation has now been slipped in (Fig. 7) to aid the argument, but this has not received a fulsome analysis. One has to start somewhere. The first discovery is often based on a single specimen or a single species (for example, the first feathered dinosaur was based on a single specimen). This is the reason that the manuscript highlights a new search image for early sponge fossils, hoping for a paradigm shift. In fact, there is already some

promising sign for additional Ediacaran-Cambrian sponges with weakly biomineralized spicules. One example is the Ediacaran Doushantuo fossils cited in the manuscript and illustrated in Figure 7; these fossils have been analyzed in great details by Muscente et al. (2015), and hence only the most important features are summarized in the current manuscript. Other examples are listed in Supplementary Table 2, with data plotted in Fig. 6. In addition, the revised manuscript cites a newly described Cambrian fossil *Cambrowania ovata*, which is also interpreted as a juvenile sponge with organic structures. In the end, we believe that our manuscript will stimulate additional paleontological, biomarker, and molecular clock studies to resolve the current controversy.

Revised text: “A positive paleontological test of this hypothesis requires the discovery of additional weakly biomineralized and aspiculate sponge fossils from the Cambrian and Precambrian. *Cambrowania ovata*, an early Cambrian fossil with organic structures reminiscent of hexactine-based spicules, has been interpreted as a possible juvenile sponge⁴⁴, and may represent another case of aspiculate or weakly biomineralized sponges in the early Cambrian.” (Line 228–232)

New text has been added to mention the importance of aspiculate calcified sponges. The fact that their exact phylogenetic positions within the Porifera are uncertain is not really relevant here. The Hetang sponge is also of uncertain phylogenetic position, so the statement that they have ‘limitingpower to resolve questions regarding to early sponge biomineralization’ is a red herring and erroneous. Why should it be accepted for one taxon (the Hetang sponge) and not many other diverse taxa (indeed whole ‘groups’ - the archaeocyaths, stromatoporoids, chaetetids and sphinctozoans). In the revised manuscript, we have removed the relevant sentence about the phylogenetic uncertainty of aspiculate calcified sponges. We would like to point out that *V. delicata* has bona-fide hexactine-based spicules that contain large proportion of organic materials and have a cylindrical axial filament. In contrast, the calcified sponges (particularly the putative Ediacaran sponge fossil *Namapoikia*) cited in the manuscript do not have spicules and thus have limited implications for the evolution of sponge spiculogenesis. Additionally, the evolution of biocalcification in *Namapoikia* requires a pre-existing organic scaffold (Wood and Penny, 2018), consistent with the hypothesis that fully biomineralized spicules were preceded by precursors of organic axial filaments that were later recruited, repeatedly and independently, for spiculogenesis.

Revised text: “Although calcified sponges—archaeocyaths, stromatoporoids, chaetetids, and spinctozoans—are notably abundant in the Paleozoic⁵⁰⁻⁵², few of them had spicules and none of them have been reported in the Precambrian, hence limiting their power to resolve questions regarding to early evolution of sponge spiculogenesis. The terminal Ediacaran fossil *Namapoikia* has been interpreted as an encrusting poriferan⁵³, but more work is needed to confirm that it is a calcified encrusting sponge

rather than a microbial structure¹⁸. In any case, *Namapoikia* is not known to have spicules either. Furthermore, the evolution of biocalcification in *Namapoikia* likely requires a pre-existing organic scaffold⁵³, consistent with the hypothesis that fully biomineralized spicules were likely preceded by organic substrates (i.e., precursors of axial filaments) that can be repeated and independently recruited as biomineralization templates and catalysts.” (Line 272–282)

Finally, how novel is this idea, and indeed these data? The organic skeleton precursor hypothesis has a long standing currency for most groups of biomineralizers. We are left then with the finding of the preservation of large axial filaments in a single species of Cambrian sponge. While this does contribute to the debate, I do not feel it is of sufficient importance to warrant publication in Nat. Comms.

Although the possibility that Precambrian animals may have been soft-bodied has been informally stated in some publications (Muscente et al., 2015), to our knowledge the hypothesis that Precambrian sponges may have had weakly biomineralized spicules or entirely organic skeletons has not been previously articulated. More importantly, the paleontological data needed to test this hypothesis have not been available or have not been discussed in the context of a transitional stage between aspiculate and fully spiculated sponges. We think this is a novel contribution of this manuscript.

Revised text: “However, emerging phylogenetic data do not require the presence of biomineralized spicules in the last common ancestor of demosponges (and that of siliceans)^{12,19,20}, prompting an alternative hypothesis that spicules may have evolved independently among sponge classes and perhaps long after the divergence of sponges²¹. This hypothesis predicts that the last common ancestor of sponges was aspiculate and that transitional forms with weakly biomineralized spicules characterize early sponges. Thus far, there have been no paleontological reports supporting this predication. In this framework, we conducted a paleontological investigation of spiculogenesis in early sponges represented by fossils from the early Cambrian (Age 2) Hetang Formation in South China (Supplementary Fig. 1). These fossils, described as *Vasispongia delicata* Tang and Xiao, n. gen. & sp. (see Supplementary Information for systematic paleontology), have siliceous spicules with large axial filaments and high organic content. In combination with a compilation of Phanerozoic sponge spicule microstructures, *V. delicata* indicates that early sponges had weakly biomineralized spicules with low fossilization potential. Together with Ediacaran microfossils that contain non-mineralized filaments possibly representing precursors of axial filaments, *V. delicata* indicates that, although sponges or sponge classes may have diverged in the Ediacaran Period or earlier, biomineralized spicules may have evolved later and independently among sponge clades.” (Line 42–58)

References

- Antcliffe, J. B., 2013, Questioning the evidence of organic compounds called sponge biomarkers: *Palaeontology*, v. 56, p. 917–925.
- Botting, J. P., and Muir, L. A., 2018, Early sponge evolution: A review and phylogenetic framework: *Palaeoworld*, v. 27, no. 1, p. 1–29.
- Budd, G. E., 2013, At the origin of animals: the revolutionary cambrian fossil record: *Current genomics*, v. 14, no. 6, p. 344–354.
- Darroch, S. A. F., Smith, E. F., Laflamme, M., and Erwin, D. H., 2018, Ediacaran extinction and Cambrian explosion: *Trends in Ecology & Evolution*, v. 33, p. 653–663.
- Dohrmann, M., and Wörheide, G., 2017, Dating early animal evolution using phylogenomic data: *Scientific Reports*, v. 7, p. 3599.
- Muscente, A. D., Michel, F. M., Dale, J. G., and Xiao, S., 2015, Assessing the veracity of Precambrian 'sponge' fossils using in situ nanoscale analytical techniques: *Precambrian Research*, v. 263, p. 142–156.
- Nettersheim, B. J., Brocks, J. J., Schwelm, A., Hope, J. M., Not, F., Lomas, M., Schmidt, C., Schiebel, R., Nowack, E. C. M., De Deckker, P., Pawlowski, J., Bowser, S. S., Bobrovskiy, I., Zonneveld, K., Kucera, M., Stuhr, M., and Hallmann, C., 2019, Putative sponge biomarkers in unicellular Rhizaria question an early rise of animals: *Nature Ecology & Evolution*, v. 3, no. 4, p. 577–581.
- Schuster, A., Vargas, S., Knapp, I. S., Pomponi, S. A., Toonen, R. J., Erpenbeck, D., and Wörheide, G., 2018, Divergence times in demosponges (Porifera): first insights from new mitogenomes and the inclusion of fossils in a birth-death clock model: *BMC Evolutionary Biology*, v. 18, p. 114.
- Wood, R., Liu, A. G., Bowyer, F., Wilby, P. R., Dunn, F. S., Kenchington, C. G., Cuthill, J. F. H., Mitchell, E. G., and Penny, A., 2019, Integrated records of environmental change and evolution challenge the Cambrian Explosion: *Nature Ecology & Evolution*, v. 3, no. 4, p. 528–538.
- Wood, R., and Penny, A., 2018, Substrate growth dynamics and biomineralization of an Ediacaran encrusting poriferan: *Proc. R. Soc. B*, v. 285, p. 20171938.
- Xiao, S., and Laflamme, M., 2009, On the eve of animal radiation: Phylogeny, ecology and evolution of the Ediacara biota: *Trends in Ecology & Evolution*, v. 24, p. 31–40.